# Rare or Overlooked Cases of Acute Acalculous Cholecystitis in Young Patients with Central Nervous System Lesion

**DOI:** 10.3390/healthcare11101378

**Published:** 2023-05-11

**Authors:** Seong-Hun Kim, Min-Gyu Lim, Jun-Sang Han, Chang-Hwan Ahn, Tae-Du Jung

**Affiliations:** 1Department of Rehabilitation Medicine, Kyungpook National University Hospital, Daegu 41944, Republic of Korea; 2Department of Rehabilitation Medicine, Kyungpook National University Chilgok Hospital, Daegu 41404, Republic of Korea; 3Department of Rehabilitation Medicine, School of Medicine, Kyungpook National University, Daegu 41944, Republic of Korea

**Keywords:** acalculous cholecystitis, young women with CNS lesion, early diagnosis

## Abstract

This case series presents two cases of acute acalculous cholecystitis (AAC)—a rare condition—in young women with central nervous system (CNS) lesions. Both patients had significant neurologic deficits and no well-known risk factors or presence of comorbidities (such as diabetes or a history of cardiovascular or cerebrovascular disease). Early diagnosis is important in cases of AAC owing to its high mortality rate; however, due to neurological deficits in our cases, accurate medical and physical examinations were limited, thereby leading to a delay in the diagnosis. The first case was of a 33-year-old woman with multiple fractures and hypovolemic shock due to a traumatic accident; she was diagnosed with hypoxic brain injury. The second case was of a 32-year-old woman with bipolar disorder and early-onset cerebellar ataxia who developed symptoms of impaired cognition and psychosis; she was later diagnosed with autoimmune encephalopathy. In the first case, the duration between symptom onset and diagnosis was 1 day, but in the second case, it was 4 days from diagnosis based on the occurrence of high fever. We emphasize that if a young woman presents with high fever, the possibility of AAC should be considered, particularly if a CNS lesion is present because it may pose difficulty in the evaluation of typical symptoms of AAC. Careful attention is thus required in such cases.

## 1. Introduction

The gallbladder is located in the right upper quadrant of the abdomen and is responsible for the production, storage, and secretion of bile. Moreover, bile is needed to digest fats in the small intestine [1]. If the bile is not properly secreted and gets accumulated, it can cause irritation and pressure in the gallbladder, thereby causing swelling and infection called cholecystitis [2]. Furthermore, cholecystitis is classified as acute or chronic depending on the duration of inflammation and the severity of symptoms [3,4]. This case series focused on acute cholecystitis (AC).

The diagnosis and severity of AC are determined using the Tokyo Guidelines 2018 (TG18), which are a set of guidelines created by a group of international experts in hepatobiliary and pancreatic surgery [5]. The guidelines include diagnostic and severity classification criteria for AC, as well as treatment guidelines based on severity. The TG18 is considered highly useful in clinical practice. First, the TG18 provides diagnostic criteria for suspected and definite AC based on local and systemic signs of inflammation and imaging findings. Local signs of inflammation include Murphy’s sign and pain, tenderness, or mass in the right upper quadrant, while systemic signs of inflammation include fever, elevated CRP, and elevated WBC count. Imaging findings characteristic of AC include gallbladder wall thickening, pericholecystic fluid, and sonographic Murphy’s sign. The TG18 designates the presence of local inflammatory signs and systemic inflammatory signs for a suspected diagnosis and requires confirmation by radiological imaging along with these two factors for a definitive diagnosis [5,6]. Second, the grading system of the TG18 categorizes patients into three grades: mild, moderate, and severe. Mild AC is defined as the presence of local inflammation without organ dysfunction or systemic inflammatory response syndrome (SIRS). Moderate AC is characterized by the presence of organ dysfunction or SIRS. Severe AC is defined by the presence of organ failure, sepsis, or septic shock and neurological dysfunction such as decreased level of consciousness [7]. Third, the TG18 also provides a set of recommendations for the management of AC based on severity grading. For mild AC, initial treatment consists of antibiotics and supportive care. For moderate and severe AC, prompt cholecystectomy is recommended [7]. However, in patients who are not candidates for surgery, percutaneous cholecystostomy or endoscopic gallbladder drainage may be considered [7,8]. It is important to evaluate the severity of comorbidities, which can be assessed by using the Charlson comorbidity index (CCI) score, and to understand the physical status of the patient, which can be evaluated by using the American Society of Anesthesiologists Physical Status (ASA-PS) classification. These tools can help determine whether to prioritize surgical or intervention treatment when making decisions about managing patients with AC [7,9,10].

Although data regarding the incidence of AC in the Korean population are unavailable, it is known that in the general population, most patients diagnosed with AC (90–95%) have gallstones (calculous cholecystitis) at diagnosis [11,12]. However, AC without gallstones (acute acalculous cholecystitis (AAC)) is an unusual condition occurring in only 5–10% of patients with AC [13]. This condition is more common in patients with critical illnesses, such as those with trauma, sepsis, severe burns, or total parenteral nutrition, who present with risk factors, such as bile stasis, gallbladder ischemia, cystic duct obstruction, and systemic illnesses [14]. Moreover, mortality due to AAC is higher (mean, 30%; range, 10–90%) than that due to calculous cholecystitis (1%) [15]. This finding can be attributed to patient conditions as well as the clinical potential of AAC to develop into pyogenic cholecystitis, which may further lead to gallbladder necrosis, perforation, abscess, or even peritonitis in many cases [16].

In previous studies, five-risk factors have been reported to be closely linked to other factors. These risk factors are as follows: sex, men; age, >60 years; comorbidities, such as diabetes mellitus; history of cardiovascular disease; and history of cerebrovascular accident, such as stroke. The characteristics of these risk factors indicate that incidence increases with age, which is also a risk factor for AC [17,18,19,20,21]. In particular, old age (>60 years) is the most important and well-known demographic risk factor [22].

Moreover, as the only risk factor, in the relevant literature, another study reported the incidence of AC in patients with stroke [23]. Among these patients, AC was observed in 28 of 2699 cases (1.04%), and the incidence of AAC was exceptionally high (14.3%). Moreover, owing to the late diagnosis, AAC manifests as severely advanced condition [23]. This can be attributed to the fact that stroke patients with significant neurologic deficits, including altered mental status, global aphasia, and severe dysarthria, have difficulty in exhibiting the classical symptoms of AC [24]. Therefore, the diagnosis of AC may be delayed in such patients, leading to a higher mortality. Furthermore, as reported earlier, the cases of AAC have even worse prognosis. Hence, early diagnosis and treatment can lead to an improvement in prognosis and reduce the risk of death due to delayed diagnosis [24,25].

Therefore, we present two cases of AAC, which can cause a high risk of mortality if diagnosis is delayed, in young women with central nervous system (CNS) lesions who did not present the abovementioned risk factors and had major neurological deficits that interfered with early diagnosis, such as stroke.

## 2. Case Presentation

### 2.1. Case 1

A 33-year-old woman (weight, 62 kg; height, 1.60 m; body mass index (BMI), 24.21 kg/m^2^, indicating overweight) presented with an independent gait and ability to perform activities of daily living (ADL), a modified Rankin score (mRS) of 0, and no specific past medical history. After multiple fractures and hypovolemic shock caused by a traumatic accident, she was transferred to our hospital’s emergency room for further management in the restoration of spontaneous circulation state.

On admission, her mental state was as follows: stupor and Glasgow Coma Scale (GCS) score of 7, with Berg Balance Scale (BBS) score of 0, Modified Barthel Index (MBI) score of 0, Mini-Mental Statement Examination (MMSE) score of 0, and bed-ridden state mRS of 5 (Table 1). Furthermore, brain magnetic resonance imaging (MRI) and single-photon emission computed tomography scans revealed hypoxic brain injury (Figure 1).

On hospital day 46, at the time of transfer to rehabilitation medicine, she presented with the following findings: alert mental state; GCS score of 15, with BBS = 0, MBI = 0, MMSE = 11; and wheelchair-bounded ADL with maximal support state mRS of 5. Furthermore, for the first time, an oral diet was started by providing a soft diet after the Gugging Swallowing Screen test. Previously, she was fed via an L-tube feeding with intermittent partial parenteral nutrition (PPN) for 46 days (Table 1).

On hospital day 65, she was mentally alert. The GCS score was 15, BBS score was 3, MBI was 15, and MMSE score was 17. She was wheelchair-bound and could perform ADL with moderate support. The mRS score was 4. Mild fever (37.2 °C), nausea, and epigastric discomfort were also noted. Moreover, her physical examination revealed a mildly distended soft abdomen with normal bowel sound, but there was a remarkable right lower quadrant pain with positive Murphy’s sign. Her blood biochemistry results were substantially normal: total bilirubin, 1.12 µmol/L; partate aminotransferase/alkaline phosphatase (AST/ALT), 45/49 U/L; amylase, 39 U/L; and lipase, 13 U/L, except for white blood cells (WBCs), 33.06/L (neutrophil 30.74/L, 93%), and C-reactive protein (CRP), 2 mg/L (Table 2). Computed tomography (CT) imaging of the abdomen and pelvis and ultrasonography were performed owing to suspicion of acute appendicitis, acute nephritis, and AC, and the results revealed significant AAC associated with a markedly dilated gallbladder (93 × 37 mm), thickened gallbladder wall, and sludge in the gallbladder (Figure 2).

The first step in determining the appropriate treatment method for the patient was to classify the severity of their case using the TG18 severity assessment and management guidelines. The patient had an elevated WBC count of 33,060/mm^3^ (>18,000/mm^3^), which was confirmed to be in a Grade 2 (moderate) state of AC (Table 3) [5,6]. Additionally, the patient’s general condition was evaluated to determine the most suitable approach for treatment. The CCI score, which was calculated as 3 points, was based on 1 point for cerebrovascular disease and 2 points for hemiplegia [9]. Furthermore, the patient also had recent (<3 months) CVA, resulting in an ASA-PS classification of IV [10]. Due to the high risk associated with performing early/urgent surgery on this patient, an initial PTGBD insertion was conducted to facilitate early GB drainage, and a delayed laparoscopic cholecystectomy was scheduled for a later time (Table 3) [7,8,26,27,28].

### 2.2. Case 2

A 32-year-old woman (weight, 33.4 kg; height, 1.58 m; BMI, 13.97 kg/m^2^, indicating underweight) was diagnosed with bipolar disorder in 2002 and early-onset cerebellar ataxia in 2010. Hence, she had a gait disorder; however, she worked as a social worker and presented with independent gait and ability to perform ADL (mRS = 2). However, since 1 month, she gradually developed impaired cognition, psychosis, and intermittent dyskinesia symptoms, including abnormal behavior. Eventually, she was admitted to the neurology department from the emergency room of our hospital for fever (temperature, 40.2 °C) and confused mental state. Aspiration pneumonia and sepsis were diagnosed in the emergency room, and intubation and mechanical ventilator care were provided. 

At admission, her mental state was as follows: drowsy, GCS of 11, with BBS = 0, MBI = 0, MMSE = 0, and bed-ridden state mRS of 5 (Table 1). During the evaluation, oromandibular automatism and seizure were observed several times. Blood biochemistry revealed anti-YO autoantibody (+, border-line) and anti-nuclear Ab titer 1:40 (+). Moreover, in the cerebrospinal fluid test, the WBC count was 1 (P100), red blood cell count was 0, protein level was 19.5, glucose level was 87 (serum, 136), and adenosine deaminase level was <1, which confirmed our diagnosis. Moreover, virus infections (herpes simplex virus infection, cytomegalovirus infection, Epstein–Barr virus infection, Enterovirus 71 infection, Japanese encephalitis, and mycoplasma), bacterial infections (listeria, tuberculosis, and neurosyphilis), and other infections (progressive multifocal leukoencephalopathy, John Cunningham virus infection, human immunodeficiency virus infection, Creutzfeldt–Jakob disease, and fungal infections) were negative in polymerase chain reaction and culture tests, and no specific findings were found in toxic encephalopathy and tumor marker tests. Furthermore, all antibody tests for autoimmune encephalitis, including anti-NMDAR (NR1), were negative. In the waking/video-electroencephalography monitoring test, status epilepticus, temporal lobe epilepsy, and moderate diffuse cerebral dysfunction were confirmed based on frequent regional spike waves on the bilateral temporal area (T7) and continuous generalized theta–delta activity. However, the brain MRI scan revealed remarkable isolated cerebellar atrophy and high signal intensity on bilateral mesial temporal and occipitoparietal lobes, indicating autoimmune encephalopathy (Figure 3).

On hospital day 104, at the time of transfer to rehabilitation medicine, her mental state was as follows: drowsiness with confusion; BBS = 0, MBI = 0, MMSE = 0; and bed-ridden and totally dependent ADL state mRS of 5. Subsequently, she was fed via L-tube feeding with PPN for 104 days (Table 1). She received intravenous antibiotics owing to the suspicion of aspiration pneumonia for consistent low-grade and intermittent high fever when she was hospitalized in the neurology department. However, no improvement was found in high fever and systemic inflammation.

On hospital day 107, she presented with fever (temperature, up to 39 °C). Because of a confused mentality and poor cooperation, physical evaluation, including detection of Murphy’s sign, was limited. Moreover, although it was impossible for her to accurately communicate, when the right upper quadrant area was pressed, her facial expression indicated pain. However, blood biochemistry results were essentially normal: WBC, 7.65/L (neutrophil 4.59/L, 59.9%); AST/ALT, 19/17 U/L; T-bil, 0.24 µmol/L, T-chol, 156; HDL, 29; and LDL, 106, except for CRP, 6.23, and triglyceride, 170 (Table 2). Furthermore, owing to the suspicion of AC with fever, abdomen–pelvis CT and ultrasonography imaging were performed, and the results revealed significant AAC with a mildly dilated gallbladder (78 × 49 mm), irregular wall thickening, and sludge in the gallbladder (Figure 4).

As in Case #1, the severity of this case was first classified using the TG18 severity assessment and management guidelines to determine the appropriate treatment method. The patient’s neurological dysfunction was confirmed to be in a Grade 3 (severe) state (Table 3) [5,6]. In addition, when evaluating the patient’s general condition to determine the treatment method, the aforementioned neurological dysfunction was considered a negative predictive factor. The patient’s CCI score was 3, with 1 point for cerebrovascular disease and 2 points for hemiplegia [9]. The patient also had recent (<3 months) CVA and sepsis, resulting in an ASA-PS classification of IV [10]. Due to the high risk associated with performing early/urgent surgery on this patient, PTGBD insertion was performed first for early GB drainage, and the patient’s current poor performance state was taken into consideration in the plan for observation (Table 3) [7,8,26,27,28].

## 3. Discussion

In this series, we presented the cases of AAC that occurred even in the absence of the five major risk factors, and early diagnosis was delayed because communication and accurate physical evaluation were restricted due to CNS [24].

We need to consider two issues. First, why AAC occurs in young women with CNS lesions, and second, why diagnosis was more delayed in the second case than in the first case.

Why ACC occurs in young women with CNS lesions?

AAC in young women with CNS lesions can be caused by various factors. Recently, many studies have reported risk factors influencing the occurrence of AAC other than major risk factors: sex, men; age, >60 years; comorbidities, such as diabetes mellitus; history of cardiovascular disease; and history of cerebrovascular accident, such as stroke [17,18,19,20].

The first factor is the link between the fasting period and incidence of AAC. Studies on the link between the fasting period and incidence of AC, including AAC, have been reported on the following two subjects. First, there is a study on the link between the mean initial consecutive fasting period and the incidence of AC, which was performed in a group of patients with aneurysmal subarachnoid hemorrhage, and it reported that the initial consecutive fasting period of 5.38 ± 2.78 days was linked to the incidence of AC [29]. Second, there are studies on the link between the mean fasting period and the incidence of AC, which were conducted in older adults with stroke, and they revealed that the mean fasting period of 15.85 ± 2.85 days was linked to the incidence of AC [30,31]. The initial consecutive fasting time was 9 days due to surgical treatment for multiple traumas in the first case and 4 days for septic shock and seizure examination and management in the second case. Moreover, the total fasting period was 16 days in the first case and >30 days in the second case due to repeated paralytic ileus and seizures (Table 4). The initial consecutive fasting time and total fasting period can be deemed as risk factors for the occurrence of AAC based on the reference value presented in the study on the link of each factor with the incidence of AC (Table 4). The mechanism by which the fasting period plays a role of risk factor for the onset of AC is as follows. Food containing proteins and long-chain triglycerides enters the gastrointestinal tract and induces cholecystokinin (CCK) secretion, which further contracts the gallbladder [32]. Hence, when the stimulation of the gastrointestinal tract is removed due to fasting, CCK secretion is reduced, causing contractile dysfunction of the gallbladder and eventually bile stasis [32]. This can induce gallbladder mucosal inflammation, which further causes AAC [32].

The second factor is autonomic dysfunction. One study reported a high risk of AC development in patients with acute stroke based on the clinical evidence of autonomic dysfunction [33]. The mechanism underlying this was sympathetic stimulation, which caused increased catecholamine secretion and vagal paresis due to significant damage to the brain center in addition to the fasting period and resulted in the hypertonicity of the sphincter of Oddi [33,34]. Notably, the muscular valve regulates the flow of bile and pancreatic juice into the duodenum, and its dysfunction can lead to bile stasis, which can ultimately cause AC [35]. As patients in the aforementioned cases also had damage in the brain center and a paralytic ileus as manifestations of autonomic dysfunction, it can be inferred that dysfunction of the sphincter of Oddi due to autonomic dysfunction led to the development of AAC.

The third factor is direct ischemic and reperfusion damage on the gallbladder. According to the studies on acute stroke as a risk factor for AC occurrence, patients with stroke frequently have an atherosclerotic risk factor that can cause ischemic and reperfusion injury of the gallbladder wall [16,19,22]. Our patients also had factors that may have induced such a reperfusion injury of the gallbladder wall. In the first case, cardiac arrest and hypoxic brain injury were caused by hypovolemic shock after multiple fractures, and in the second case, there was recurrent aspiration and a history of septic shock. The abovementioned cases had the risk of ischemic and reperfusion of the gallbladder wall due to the associated diseases.

Why diagnosis was more delayed in the second case than in the first case?

In the first case, the duration between symptom onset and diagnosis was 1 day, but in the second case, the time of symptom onset could not be specified, and it took approximately 4 days from the diagnosis based on the occurrence of high fever.

Compared with Case 1, two reasons can be deduced for the delay in AAC diagnosis in Case 2. First, the underlying medical condition was different in both cases. The patient in Case 2 had repeated low-grade and intermittent high fever due to aspiration pneumonia after hospitalization; hence, it took 4 days to suspect another cause of fever and evaluate it. Second, the reporting of symptoms differs according to the patient’s level of consciousness and limitations of physical examination due to impaired cognitive level. In Case 1, the patient was alert with moderately impaired cognition. Although there was cognitive decline, communication was possible; moreover, physical examination, including Murphy’s sign test, was possible. However, the mental state of the patient in Case 2 was weak with severely impaired cognition, suggesting a severely impaired cognition state, indicating that she could not communicate or undergo an adequate physical examination.

## 4. Conclusions

In contrast to the findings of previous reports, we reported that AAC can occur in young individuals; therefore, it is essential to consider the possibility of AAC when the patient is presented with high fever. In particular, patients with CNS lesions should be treated with caution because they frequently present with severe neurological issues, such as changes in mental state and cognitive impairment, which can make it difficult for them to exhibit typical AAC symptoms.

## Figures and Tables

**Figure 1 healthcare-11-01378-f001:**
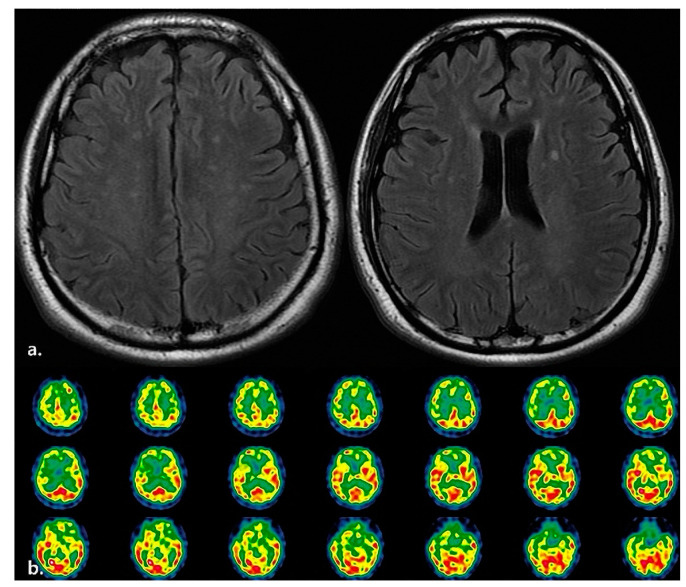
Case #1. (**a**) Brain MRI (T2/FLAIR). Multiple high signal intensity on bilateral cerebral hemisphere. (**b**) Brain SPECT. Markedly decreased perfusion in both frontal cortex and right basal ganglia.

**Figure 2 healthcare-11-01378-f002:**
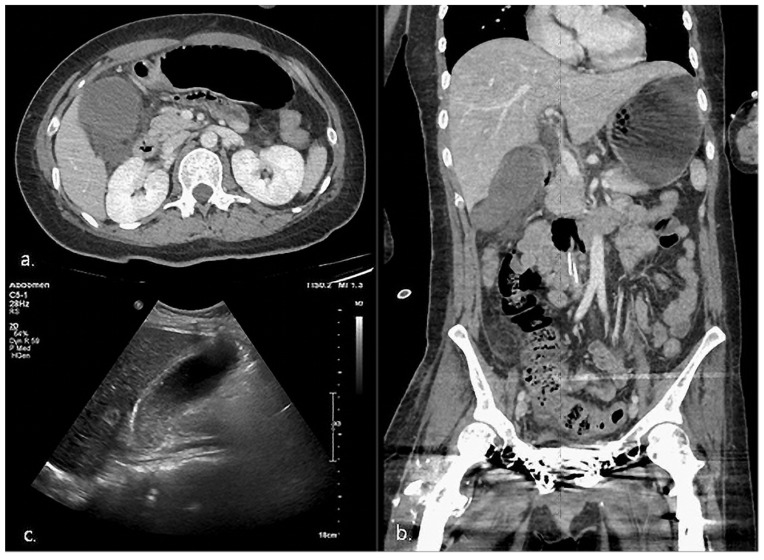
Case #1. (**a**,**b**) Abdomen-pelvic CT. the gallbladder (GB) is significantly distended, and a small amount of fluid is present in the right inferior perihepatic space, including the GB fossa. (**c**) Hepatobiliary US. Showing thickening of the GB wall, and the presence of amorphous echogenic sludge, considered to be a biliary sludge, inside the gallbladder.

**Figure 3 healthcare-11-01378-f003:**
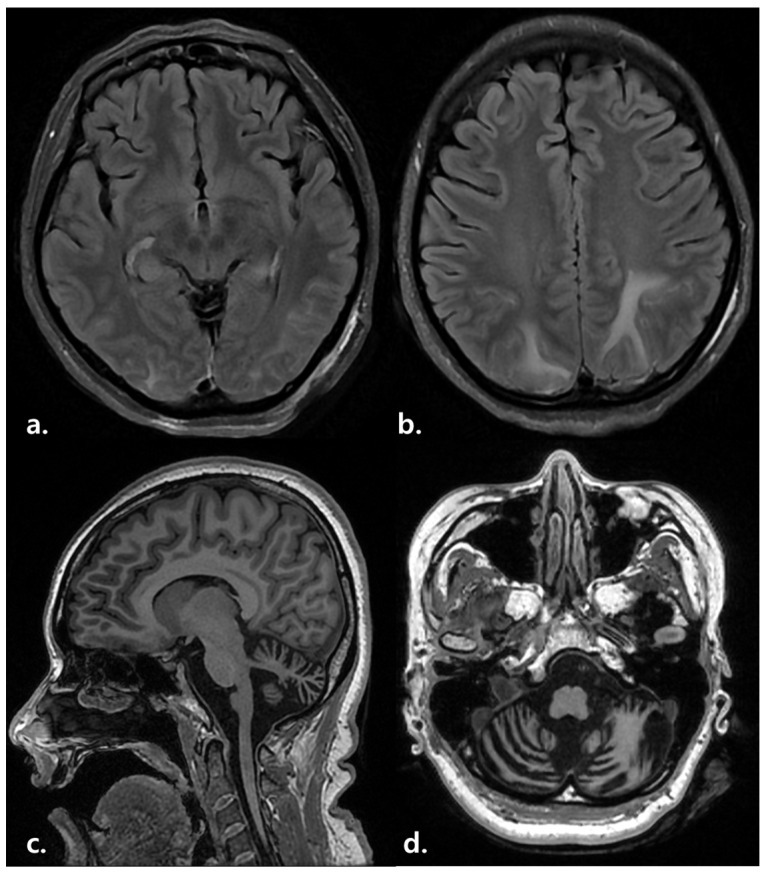
Case #2. (**a**,**b**) Brain MRI (T2/FLAIR). High signal intensity on bilateral mesial temporal and occipitoparietal lobes. (**c**,**d**) Brain MRI (T1). Significant isolated cerebellar atrophy.

**Figure 4 healthcare-11-01378-f004:**
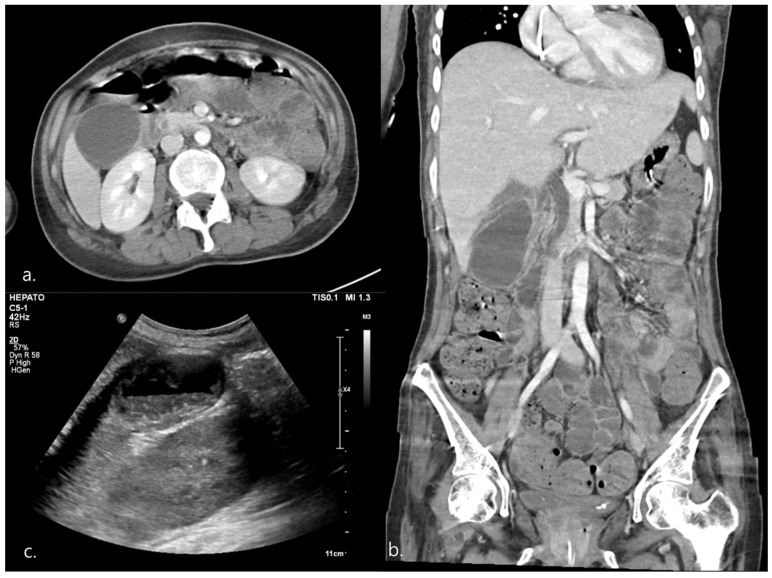
Case #2. (**a**,**b**) Abdomen-pelvic CT. the gallbladder (GB) is distended, and has a markedly diffuse-irregular wall thickening. (**c**) Hepatobiliary US. Showing the presence of biliary sludge inside the GB.

**Table 1 healthcare-11-01378-t001:** Characteristics of two cases.

Case	Age (yrs)	Sex	Diagnosis of CNS Lesion	Time	BBS	MBI	MMSE	mRS	Diet
1	33	F	Hypoxic brain injury	Admission (15 August 2021)	0	0	0	5	L-tube
				RM transfer (30 September 2021)	0	0	11	5	Soft diet
				AC diagnosis (20 October 2021)	3	5	17	4	Regular diet
2	32	F	Autoimmune encephalitis	Admission (29 January 2021)	0	0	0	5	L-tube
				RM transfer (13 May 2021)	0	0	0	5	L-tube
				AC diagnosis (15 May 2021)	0	0	0	5	L-tube

BBS, Berg Balance Scale; MBI, Modified Barthel Index; MMSE, Mini-Mental State Examination; mRS, modified Rankin scale; RM, rehabilitation medicine; AC, acute cholecystitis.

**Table 2 healthcare-11-01378-t002:** Diagnostic course according to the Tokyo Guidelines 2018 (TG18).

Case	Time to Symptom Onset (Days)	Time to Diagnosis (Days)	Symptom		Image Findings					Diagnosis
			Local	Systemic	Diagnostic Tool	GB Size (mm)	Wall Thickness (mm)	Biliary Sludge	GB Stone	
1	HD 65	1	Epigastric discomfort RLQ tenderness Murphy’s sign *	Fever * Elevated CRP *, WBC *	CT, US	93 × 41 *	3.76	(+) *	(−)	Definite AC
2	HD 105	4	RUQ tenderness *	Fever Elevated CRP *	CT, US	78 × 49	4.1 *	(+) *	(−)	Definite AC

* Factors correspond to the TG18 diagnostic criteria for acute cholecystitis (AC). HD, hospital day; RLQ, right lower abdominal quadrant; RUQ, right upper abdominal quadrant; CRP, C-reactive protein; WBC, white blood cell; US, ultrasonography; CT, computed tomography.

**Table 3 healthcare-11-01378-t003:** Severity assessment and management according to the Tokyo Guidelines 2018 (TG18).

**Case**	Diagnosis	Severity	General Condition		Plan
			CCI Score [9]	ASA-PS Classification [10]	
1	Definite AC	Grade II (Moderate)	Total score 3	ASA IV	1st. Early GB drainage (PTGBD) 2nd. Delayed/elective LC
2	Definite AC	Grade III (Severe)	Total score 3	ASA IV	1st. Early GB drainage (PTGBD) 2nd. Observation

ASA-PS, American Society of Anesthesiologists Physical Status; CCI, Charlson comorbidity index; GB, gallbladder; PTGBD, percutaneous transhepatic gallbladder drainage; LC, laparoscopic cholecystectomy.

**Table 4 healthcare-11-01378-t004:** Fasting period.

Case	The Initial Consecutive Fasting Time (Days)	The Total Fasting Time (Days)
1	9	16
2	4	>30
Reference value	5.38 ± 2.78 *	15.85 ± 2.85 **

* The mean initial consecutive fasting period reported in the aneurysmal subarachnoid hemorrhage group. ** The mean fasting period reported in the elderly patients with stroke group.

## Data Availability

The data presented in this study are available on request from the corresponding author. The data are not publicly available due to patient privacy.

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
