# Peer review of "Rare or Overlooked Cases of Acute Acalculous Cholecystitis in Young Patients with Central Nervous System Lesion"

_healthcare, 2023, doi:10.3390/healthcare11101378_

Round 1

Reviewer 1 Report

good job, After carefully reviewing the file I have several important modifications to make: - Set an objective with said work, since you always have to reach a conclusion through a marking, the reason for said case. - the tables are not placed in gold of appearance in the text. - Table 3 would not put it into the discussion, it would be his thing to be in the upper section. the tamebin figures are misplaced depending on the appearance of the information in the text. - What does the d of the tables mean? (Table 3) so I find it very interesting.

Author Response

  1. good job, After carefully reviewing the file I have several important modifications to make: - Set an objective with said work, since you always have to reach a conclusion through a marking, the reason for said case. - the tables are not placed in gold of appearance in the text. - Table 3 would not put it into the discussion, it would be his thing to be in the upper section. the tamebin figures are misplaced depending on the appearance of the information in the text. - What does the d of the tables mean? (Table 3) so I find it very interesting.

Response: We appreciate your valuable comment. We've made some changes to the positioning of the tables and figures in accordance with your valuable input. On top of that, we've added an extra table to make the case report more comprehensive, and we've renamed the previous Table 3 to Table 4. It's worth noting that the abbreviation "d" used in the original Table 3 to denote days could have been misinterpreted and caused confusion, so we've replaced it with the full word "days." Managing the records of two patients in the case report has proven to be challenging, and organizing their data effectively in each table has become difficult. Furthermore, Table 4, which presents the details of the fasting period, is considered the main topic of discussion and has been placed before the main discussion. We really appreciate you taking the time to read this report, especially considering your busy schedule. Thank you again for your valuable feedback.

Reviewer 2 Report

The paper presents two cases of acute acalculous cholecystitis in young patients with severe neurological deficits.

The article is well-written and well-argued. The topic is interesting and can improve a subject not covered much in the literature.

In my opinion, there are some minor revision to be made, and to update the literature used by the authors a little:

 1) In the Introduction the definition of acute cholecystitis is mentioned, I think it is important to report the definition of TG18 widely used in the world of surgery.

Tokyo Guidelines 2018: diagnostic criteria and severity grading of acute cholecystitis (with videos). Masamichi Yokoe, Jiro Hata, Tadahiro Takada, Steven M. Strasberg, Horacio J. Asbun, Go Wakabayashi, Kazuto Kozaka, Itaru Endo, Daniel J. Deziel, Fumihiko Miura, Kohji Okamoto … et al. doi.org/10.1002/jhbp.515. Journal of Hepato-Biliary-Pancreatic Sciences (2018) 25(1); 41-54.

2) In the introduction some risk factors are defined, I think it is important also to mention more recent and surgical articles in order to give a more complete picture of the problem, cite the following articles or help yourself by reading them to cite more recent articles.:

Cholecystectomy for acute cholecystitis in octogenarians: Impact of advanced age on postoperative outcome. Vaccari S, Lauro A, Cervellera M, et al. Minerva Chirurgica, 2019, 74(4), pp. 289–296

Early versus delayed approach in cholecystectomy after admission to an emergency department. A multicenter retrospective study. Vaccari S, Lauro A, Cervellera M, et al. Giornale di Chirurgia, 2018, 39(4), pp. 232–238

Age matters: A study of clinical and economic outcomes following cholecystectomy in elderly Americans. Kuy, S., Sosa, J.A., Roman, S.A., Desai, R., Rosenthal, R.A. (2011) American Journal of Surgery, 201 (6), pp. 789-796. Cited 61 times. doi: 10.1016/j.amjsurg.2010.04.018

Gallbladder perforation: morbidity, mortality and preoperative risk prediction. Ausania, F., Guzman Suarez, S., Alvarez Garcia, H., Senra del Rio, P., Casal Nuñez, E. (2015) Surgical Endoscopy, 29 (4), pp. 955-960. doi: 10.1007/s00464-014-3765-6

3) The treatment of both cases was PTGBD. Why this indication? Why didn't you think about surgery? Motivate your choices on the basis of the literature. For example you can cite the following paper

Tokyo Guidelines 2018: flowchart for the management of acute cholecystitis. Kohji Okamoto, Kenji Suzuki, Tadahiro Takada, Steven M. Strasberg, Horacio J. Asbun, Itaru Endo, Yukio Iwashita, Taizo Hibi, Henry A. Pitt, Akiko Umezawa, Koji Asai, Ho-Seong Han et al. doi.org/10.1002/jhbp.516 Journal of Hepato-Biliary-Pancreatic Sciences (2018) 25(1); 55-72.

4) I would add a table with data extracted from the literature regarding acute acalculous cholecystitis. Try to summarize the cases in the literature similar to yours, perhaps also indicating the treatment.

5) Finally, my curiosity. Have the patients resolved their acute illnesses? Are they alive?

Author Response

Response to Reviewer’s Comments

Reviewer #2

  1. The paper presents two cases of acute acalculous cholecystitis in young patients with severe neurological deficits. The article is well-written and well-argued. The topic is interesting and can improve a subject not covered much in the literature. In my opinion, there are some minor revision to be made, and to update the literature used by the authors a little:
  • In the Introduction the definition of acute cholecystitis is mentioned, I think it is important to report the definition of TG18 widely used in the world of surgery.

Response: We appreciate your valuable comment. We've made an addition to the introduction [38~65 lines] of the case report by including a new section that introduces the Tokyo Guidelines 2018 (TG 18). This was done to enhance the report's comprehensiveness, and we were able to do so by making reference to the articles that you kindly shared with us. You'll find the added content below. Please see the attachment

 The diagnosis and severity of AC are determined using the Tokyo Guidelines 2018 (TG 18), which are a set of guidelines created by a group of international experts in hepatobiliary and pancreatic surgery.[5] The guidelines include diagnostic and severity classification criteria for AC, as well as treatment guidelines based on severity. The TG 18 is considered highly useful in clinical practice.

First, TG18 provides diagnostic criteria for suspected and definite AC based on local and systemic signs of inflammation and imaging findings. Local signs of inflammation include Murphy's sign and pain, tenderness or mass in the right upper quadrant, while systemic signs of inflammation include fever, elevated CRP, and elevated WBC count. Imaging findings characteristic of AC include gallbladder wall thickening, pericholecystic fluid, and sonographic Murphy's sign. TG18 designates the presence of local inflammatory signs and systemic inflammatory signs for a suspected diagnosis, and requires confirmation by radiological imaging along with these two factors for a definitive diagnosis.[5,6]

Second, the grading system of TG18 categorizes patients into three grades: mild, moderate, and severe. Mild AC is defined as the presence of local inflammation without organ dysfunction or systemic inflammatory response syndrome (SIRS). Moderate AC is characterized by the presence of organ dysfunction or SIRS. Severe AC is defined by the presence of organ failure, sepsis, or septic shock, and neurological dysfunction such as decreased level of consciousness.[7]

Third, TG18 also provides a set of recommendations for the management AC based on severity grading. For mild AC, initial treatment consists of antibiotics and supportive care. For moderate and severe AC, prompt cholecystectomy is recommended.[7] However, in patients who are not candidates for surgery, percutaneous cholecystostomy or endoscopic gallbladder drainage may be considered.[7,8] It is important to evaluate the severity of comorbidities, which can be assessed by using the Charlson comorbidity index (CCI) score, and to understand the physical status of the patient, which can be evaluated by us-ing the American Society of Anesthesiologists Physical Status (ASA-PS) classification. These tools can help determine whether to prioritize surgical or intervention treatment when making decisions about managing patients with AC.[7,9,10]

  • In the introduction some risk factors are defined, I think it is important also to mention more recent and surgical articles in order to give a more complete picture of the problem, cite the following articles or help yourself by reading them to cite more recent articles.:

Response: We appreciate your valuable comment. I am grateful to have read the papers you sent over. Furthermore, I have conducted our own search and attached additional papers that are relevant to this case report, including the one you requested. Specifically, I was able to expand and enrich the content by referring to the TG18 guideline thesis. Thanks to your contribution, I am confident that the case report is now more comprehensive. For your convenience, I have provided a list of the attached papers below, with the reference numbers indicating their order in the report. Once again, thank you for your invaluable feedback. Please see the attachment

  1. Yokoe, M.; Hata, J.; Takada, T.; Strasberg, S.M.; Asbun, H.J.; Wakabayashi, G.; Kozaka, K.; Endo, I.; Deziel, D.J.; Miura, F.; et al. Tokyo Guidelines 2018: diagnostic criteria and severity grading of acute cholecystitis (with videos). J Hepatobiliary Pan-creat Sci 2018, 25, 41-54, doi:10.1002/jhbp.515.
  2. Hirota, M.; Takada, T.; Kawarada, Y.; Nimura, Y.; Miura, F.; Hirata, K.; Mayumi, T.; Yoshida, M.; Strasberg, S.; Pitt, H.; et al. Diagnostic criteria and severity assessment of acute cholecystitis: Tokyo Guidelines. J Hepatobiliary Pancreat Surg 2007, 14, 78-82, doi:10.1007/s00534-006-1159-4.
  3. Okamoto, K.; Suzuki, K.; Takada, T.; Strasberg, S.M.; Asbun, H.J.; Endo, I.; Iwashita, Y.; Hibi, T.; Pitt, H.A.; Umezawa, A.; et al. Tokyo Guidelines 2018: flowchart for the management of acute cholecystitis. J Hepatobiliary Pancreat Sci 2018, 25, 55-72, doi:10.1002/jhbp.516.
  4. Vaccari, S.; Lauro, A.; Cervellera, M.; Casella, G.; D'Andrea, V.; Di Matteo, F.M.; Santoro, A.; Panarese, A.; Gulotta, E.; Ci-rocchi, R.; et al. Early versus delayed approach in cholecystectomy after admission to an emergency department. A multi-center retrospective study. G Chir 2018, 39, 232-238.
  5. Charlson, M.E.; Pompei, P.; Ales, K.L.; MacKenzie, C.R. A new method of classifying prognostic comorbidity in longitudi-nal studies: development and validation. J Chronic Dis 1987, 40, 373-383, doi:10.1016/0021-9681(87)90171-8.
  6. Doyle, D.J.; Hendrix, J.M.; Garmon, E.H. American Society of Anesthesiologists Classification. In StatPearls; StatPearls Pub-lishing Copyright © 2023, StatPearls Publishing LLC.: Treasure Island (FL), 2023.
  7. Kuy, S.; Sosa, J.A.; Roman, S.A.; Desai, R.; Rosenthal, R.A. Age matters: a study of clinical and economic outcomes following cholecystectomy in elderly Americans. Am J Surg 2011, 201, 789-796, doi:10.1016/j.amjsurg.2010.04.018.
  8. Harris, D.A.; Sheu, E.G. Biliary Tract. In Current Diagnosis & Treatment: Surgery, 15e, Doherty, G.M., Ed.; McGraw Hill LLC: New York, NY, 2020.
  9. Akhan, O.; Akinci, D.; Ozmen, M.N. Percutaneous cholecystostomy. Eur J Radiol 2002, 43, 229-236, doi:10.1016/s0720-048x(02)00158-4.
  10. Lee, M.J.; Saini, S.; Brink, J.A.; Hahn, P.F.; Simeone, J.F.; Morrison, M.C.; Rattner, D.; Mueller, P.R. Treatment of critically ill patients with sepsis of unknown cause: value of percutaneous cholecystostomy. AJR Am J Roentgenol 1991, 156, 1163-1166, doi:10.2214/ajr.156.6.2028859.
  11. vanSonnenberg, E.; D'Agostino, H.B.; Goodacre, B.W.; Sanchez, R.B.; Casola, G. Percutaneous gallbladder puncture and cholecystostomy: results, complications, and caveats for safety. Radiology 1992, 183, 167-170, doi:10.1148/radiology.183.1.1549666.

  • The treatment of both cases was PTGBD. Why this indication? Why didn't you think about surgery? Motivate your choices on the basis of the literature. For example you can cite the following paper

Response: We appreciate your valuable comment. Thank you for providing the Tokyo Guidelines 2018 flowchart for the management of acute cholecystitis. I have incorporated the process for selecting a treatment method for the patients mentioned in the case report, which can be found on page 4, lines 124-143 and page 6, lines 196-206. Additionally, I have included Table 3, which summarizes severity assessment and management in accordance with The Tokyo Guidelines 2018 (TG18) regarding diagnosis, severity assessment, and treatment. You can find both the information and table below. Thank you for your assistance. Please see the attachment

< Case#1, page 4, lines 124-143 >

The first step in determining the appropriate treatment method for the patient was to classify the severity of their case using the TG 18 severity assessment and management guidelines. The patient had an elevated WBC count of 33,060/mm3 (>18,000/mm3), which was confirmed to be in a Grade 2 (moderate) state of AC (Table 3).[5,6] Additionally, the patient's general condition was evaluated to determine the most suitable approach for treatment. The CCI score, which was calculated as 3 points, was based on 1 point for Cerebrovascular disease and 2 points for Hemiplegia[9]. Furthermore, the patient also had recent (<3 months) CVA, resulting in an ASA-PS classification of IV[10]. Due to the high risk associated with performing early/urgent surgery on this patient, an initial PTGBD insertion was conducted to facilitate early GB drainage, and a delayed laparoscopic cholecystectomy was scheduled for a later time (Table 3).[7,8,26-28]

< Case#2, page 6, lines 196-206 >

As in Case #1, the severity of this case was first classified using the TG 18 severity assessment and management guidelines to determine the appropriate treatment method. The patient's neurological dysfunction was confirmed to be in a Grade 3 (severe) state(Table 3).[5,6] In addition, when evaluating the patient's general condition to deter-mine the treatment method, the aforementioned neurological dysfunction was considered a negative predictive factor. The patient's CCI score was 3, with 1 point for Cerebrovascular disease and 2 points for Hemiplegia.[9] The patient also had recent (<3 months) CVA and sepsis, resulting in an ASA-PS classification of IV.[10] Due to the high risk associated with performing early/urgent surgery on this patient, PTGBD insertion was performed first for early GB drainage, and the patient's current poor performance state was taken into consideration in the plan for observation(Table 3).[7,8,26-28]

  • I would add a table with data extracted from the literature regarding acute acalculous cholecystitis. Try to summarize the cases in the literature similar to yours, perhaps also indicating the treatment.

Response: We appreciate your valuable comment. I'm glad to say that I searched for papers related to the TG18 guideline you shared and reviewed the contents of acalculous cholecystitis management. It turns out that the management is done following the TG18 guideline, taking into account the severity of AC and the overall health of the patient, which is nicely summarized in review No. 3. Thank you for providing these resources. Please see the attachment

  • Finally, my curiosity. Have the patients resolved their acute illnesses? Are they alive?

Response: We appreciate your valuable comment. We appreciate your interest in our case report, despite your busy schedule. Patient case #1 demonstrated significant improvement following a combination of antibiotics, supportive care, and PTGBD insertion. The PTGBD was subsequently removed, and during a follow-up visit at the ASA gastroenterology OPD three months later, the patient's ASA-PS classification had improved. A delayed laparoscopic cholecystectomy was performed as a result. It is unfortunate to report that Patient Case #2 has been receiving treatment in an intensive care unit and requiring mechanical ventilation due to recurring sepsis, along with episodes of loss of consciousness and seizures. However, the prognosis for the patient is not looking favorable. Please see the attachment

Reviewer 3 Report

Although it is interesting case series, no novelty/ scientific evidence to be published

Author Response

  1. Although it is interesting case series, no novelty/ scientific evidence to be published

Response: We appreciate your valuable comment. We greatly appreciate your opinion. Nevertheless, the paper above sheds light on the occurrence of acute acalculous cholecystitis in patients with central nervous system (CNS) lesions, a patient group that lacks known risk factors based on prior research. This information is expected to aid clinicians who are treating these patients directly. We genuinely acknowledge and value the time and effort you have dedicated to reading this case report.
